# Optimal waist circumference cut-off points for predicting metabolic syndrome among females of reproductive age in Wakiso district, central Uganda

David Lubogo[1]*, Henry Wamani[1], Roy William Mayega[2], Christopher Garimoi Orach[1]

**1** Department of Community Health and Behavioural Sciences, School of Public Health, College of Health Sciences, Makerere University, Kampala, Uganda, **2** Department of Epidemiology and Biostatistics, School of Public Health, College of Health Sciences, Makerere University, Kampala, Uganda

\* Lubogodad@yahoo.com

## Abstract

Metabolic syndrome (MetS) poses a significant challenge to global public health, due to its strong association with Type 2 diabetes and cardiovascular disease. Waist circumference (WC) is a convenient metric for diagnosing MetS. Our study sought to establish waist circumference cut-off points that predict MetS among females of reproductive age in Wakiso district, central Uganda. The data collected were from a cross-sectional study conducted in Wakiso district, central Uganda, involving 697 randomly selected females aged 15 to 49, between 9th June 2021 and 17th August 2021. Data included MetS components: WC, High-Density Lipoprotein (HDL) Cholesterol, triglycerides, blood pressure, and fasting blood glucose. MetS status was identified based on two or more MetS components excluding WC. ROC analysis established the average optimal WC. The accuracy and performance of the cut-off points were evaluated through sensitivity, specificity, positive likelihood ratio, and the Youden Index. Of the 634 participants who were aged 18-49 years, 51.4% had two or more MetS risk factors. Their average optimal WC threshold was 80.3 cm. Variations in optimal WC thresholds were observed across different age groups: 80.4 cm (18-24 years), 79.9 cm (25-34 years), 85.6 cm (35-44 years), and 91.1 cm (45-49 years) respectively. The area under the ROC curve at the cut point for these age groups ranged from 0.78 to 0.86, indicating good discriminatory capability. The sensitivity ranged from 85% to 97%, specificity from 58% to 88%, and the Youden Index from 0.557 to 0.729. A waist circumference of 80.3 cm was the optimal threshold for identifying metabolic syndrome in females between the ages of 18 and 49 years in the setting. This finding concurs with the guidance set forth by the International Diabetes Federation. Additionally, study participants' WC cut-offs varied, ranging from 79.9 cm to 91.1 cm, depending on their age.

## Introduction

Metabolic syndrome (MetS) corresponds to the combination of several disorders closely related to abdominal obesity, [1] which is a strong predictor of adverse metabolic outcomes including insulin resistance, hypertension, and dyslipidemia [2,3].

**Data availability statement:** Data already provided as part of the submitted article.

**Funding:** DL received funding for this work from the Government of Uganda through the Makerere University Research and Innovation Fund (MakRIF) and the Strengthening Education and Training Capacity in Sexual and Reproductive Health and Rights (SET-SRHR) Project in Uganda. The funders had no role in study design, data collection and analysis, decision to publish, or preparation of the manuscript.

There are several definitions for MetS. However, according to the International Diabetes Federation (IDF), a diagnosis of MetS requires the presence of abdominal (central) obesity, measured by waist circumference, with ethnic-specific thresholds of ≥ 94 cm and ≥ 90 cm for European and Asian men, respectively, and ≥ 80 cm for women of all ethnic groups, and at least two of the following four factors, reduced HDL Cholesterol, elevated triglycerides, high blood pressure, and elevated fasting blood glucose [4–6].

Metabolic syndrome is a significant global public health challenge, contributing to high morbidity and mortality, with its prevalence increasing in Africa [7–10]. The failure to address MetS significantly increases the chances of developing Type 2 diabetes and cardiovascular disease [11–13]. Therefore, early identification of cases and management are key in preventing disease progression and the associated long-term health consequences [14].

The measurement of waist circumference (WC) serves as a simple and practical indicator for assessing abdominal obesity, which is a key component of metabolic syndrome [15,16]. The World Health Organization (WHO) and other scientific organizations strongly recommend using waist circumference measurement in screening, diagnosis, and routine clinical care to guide patient management [5,17]. This requires an acceptable waist circumference cut-off for risk stratification. However, establishing universally accepted WC cut-offs has been challenging due to differences in abdominal tissue composition among ethnic groups [18]. This is because genetics, environmental, and lifestyle elements impact abdominal obesity [19] and overall body composition [20].

Waist circumference is a simple, useful and superior [21–23] predictor for MetS compared to body mass index. Its simplicity and ability to capture abdominal obesity make it a widely used tool [24] in both clinical and public health practice. Abdominal adipose tissue positively correlates with waist circumference and waist-hip ratio and is associated with several metabolic abnormalities including adverse lipid profiles, decreased insulin sensitivity, and decreased glucose tolerance all of which are risk factors for T2DM and CVD.

The current waist circumference cut-off points for predicting metabolic syndrome vary by population and guidelines. For instance, according to the International Diabetes Federation (IDF) guidelines, the waist circumference cut-off for men is ≥ 94 cm for Europeans and ≥ 90 cm for South Asians, Chinese and Japanese. For women, the waist circumference is ≥80 cm for all ethnicities [24]. Sub-Saharan Africans use European waist circumference cut-offs [24]. On the other hand, the WHO Expert Consultation (2008) cut-off points for men are ≥ 94 cm and ≥80 cm for women [20].

Metabolic syndrome is more prevalent among females than males [8,11], directly affects their reproductive roles, and contributes to adverse maternal [25] and perinatal [26,27] outcomes.

Women of reproductive age (15-49 years) experience distinct physiological and hormonal changes that influence their health and metabolic health [28]. Hormonal fluctuations due to the menstrual cycle, pregnancy, and menopause, as well as reproductive health conditions such as polycystic ovary diseases (PCOS), can alter body fat distribution [29], leading to abdominal obesity. This abdominal obesity, along with its associated metabolic abnormalities, can directly affect women's reproductive health [30]. Obesity in childhood can also lead to the development of metabolic diseases such as diabetes or cardiovascular disease later in life [31].

Uganda and other countries in sub-Saharan Africa use WC cut-offs endorsed by international scientific organizations including the IDF and WHO [4,20]. Nevertheless, it is still unclear whether established thresholds, such as the ≥ 80 cm suggested for females, are valid and appropriate for the African population. Hence, it the necessary to conduct research on the diverse African population groupings including Uganda and to establish sex-specific WC cut-offs as recommended by WHO and others [5,20].

To date, most studies on optimal waist circumference cut-off points have been conducted in high-income countries. In low-income countries such as Uganda, the optimal ethnic WC cut-off has not been established particularly for females of reproductive age. Given the unique metabolic challenges faced by women of reproductive age, determining optimal WC cut-off points that are sensitive to their specific physiological context is warranted to improve early detection and management of the condition.

Therefore, this study aimed at determining the average waist circumference and age-specific cut-off points that are optimal for predicting MetS among females of reproductive age in Wakiso district, central Uganda. This may assist in the early detection, prevention and management of metabolic syndrome. The hypothesis for the study was that the optimal waist circumference for females aged 15-49 years might not be equal to 80.0 cm.

## Methods

### Study design

We carried out a cross-section study in Wakiso district between 9th June 2021 and 17th August 2021 using a random sample of female participants from 15 to 49 years of age. This study used pre-weighted data, where sampling weights were applied to adjust for selection probabilities at different levels (enumeration areas, household size, and individuals) and response rates [32]. This ensured the representativeness of the target population.

### Study population

The study population included females aged 15-49 years residing in Wakiso district, central Uganda. Participants were randomly selected from urban and rural communities to ensure diversity. Eligible participants had resided in Wakiso for at least a year and were identified from a database of 800 individuals, who had anthropometric, physical, and biochemical tests carried out on them to determine the prevalence of MetS in the district.

We excluded participants who were pregnant, those within six weeks after childbirth, alcoholics, smokers, those with chronic diseases such as diabetes mellitus, hypertension, hyperlipidaemia, allergic disorders, depression, and those with known carrier states of hepatitis B, hepatitis C, or human Immunodeficiency virus.

### Study setting

The study was conducted in Wakiso district, situated in central Uganda, with an estimated total population of 3,105,700, based on the 2014 census. More than half (52.7%) of the residents are females. Of these, 793,770 are in the reproductive age bracket of 15-49 years [33]. The district comprises two counties, namely Kyaddondo and Busiro, and four Municipal Councils - Entebbe, Nansana, Kira, and Makindye-Ssabagabo that together constitute eight sub-counties, eight town councils, 148 parishes, and 704 villages. The Uganda Bureau of Statistics provided the sampling frame for the study, and it included 32 randomly selected enumeration areas in the district. Twenty-eight (28) households were sampled per enumeration area.

### Sample size calculation

The sample size was computed using the OpenEpi, Version 3.01, open-source calculator for a proportion or descriptive study [34]. Using a population size for a finite population of 1,000,000, confidence limits of 5%, a design effect of 2.0 and a prevalence (p) of 40.2% +/-5 obtained from a study conducted among females in an urban population in Kenya to

determine MetS prevalence [35]. A sample size of 739 was obtained; after adjusting for a response rate of 90%, the sample size was 813. The final sample used in the study was 800 respondents. We estimated a MetS prevalence of 15% and expected 120 respondents to be with MetS out of the total sample size of 800. This sample was considered appropriate for assessing the sensitivity and specificity of WC as a predictor for MetS in this sample.

## Measurements and variables

We used the population sample data set from a prevalence study to obtain participant data on anthropometric measurements (waist circumference), physical measurements (blood pressure), and biochemical measurements (fasting plasma glucose, triglycerides, and low HDL Cholesterol) collected as per the WHO STEP wise approach [36].

Quality control measures were undertaken to ensure the data for the prevalence study were of quality. They included the employment of well-trained data collectors/research assistants (female nurses and graduate nutritionists) who were knowledgeable of the study protocol and had skills in using the Open Data Kit (ODK) for data collection. The data collection tools were field tested from a district near the study district. The data collected in the field were entered in the ODK tool embedded with skip patterns and constraints to ensure data quality and reduce bias. Quantitative data (ODK data) was downloaded from the server and exported to Stata version 14 for cleaning. This ensured reasonable quality control and data management for this study.

## Study variables

**Dependent variable.** Metabolic syndrome was assessed by determining if an individual had three or more of the five components namely: low HDL cholesterol (<50 mg/dL (1.3 mmol/l) in females) or treatment for it, high triglycerides (≥150 mg/dL (1.7 mmol/L) or treatment for it, elevated fasting blood glucose (≥ 100 mg/dL (5.6 mmol/l) or diabetes diagnosis, or diabetes treatment, high blood pressure (≥130/ ≥85 mmHg) or treatment for it, and increased waist circumference. These components were derived from the Joint Interim Statement (JIS) [5]. This study assessed MetS using two of the four metabolic components, excluding waist circumference. This approach was used to evaluate the independent contribution of waist circumference to metabolic risk. Similar methods have been applied in previous studies [37–39].

Detailed information on the measurements of the variables shown in the baseline characteristics can be found in Lubogo et al.(2024) [32].

## Data analysis

Data were analysed using Stata statistical software (SE/14.0, StatCorp, College Station, TX, USA [40]. Descriptive statistics were employed to summarise participant characteristics. Data on continuous variables were summarised using the independent sample T-test and categorical variables were analysed using the Chi-square test. The results were presented as means with standard deviation and frequencies/percentages.

Waist circumference cut-off points for various age groups (18-24, 25-34, 35-44, 45-49, 15-49, and 18-49, years) were evaluated using sensitivity and specificity techniques to identify the most effective predictive values for MetS within the study age groups.

The research investigated the Area under the Curve (AUC) for different components of MetS such as elevated fasting blood glucose, high triglycerides, high blood pressure and low HDL cholesterol. To conduct this analysis, an ROC curve was plotted for the association between waist circumference (WC) and two or more components of MetS.

The optimal cut-off points for WC, which achieved maximum accuracy, were determined by plotting sensitivity and (1-specificity) on a graph.

The study employed the Youden Index method [41] expressed as [Maximum (sensitivity + specificity - 1)] to ascertain the optimal WC values and also calculated the positive predictive values (PPV) and negative predictive values (NPV).

## Ethical considerations

We obtained approval from the Higher Degrees Research and Ethics Committee of the School of Public Health, College of Health Sciences, Makerere University (Protocol number: MakSPH-REC, 071) and the National Council for Science and Technology of Uganda (Registration number: UNCST, HS1281ES). All study participants provided written informed consent and were assured of the confidentiality of their information. Written informed consent was obtained from parents or guardians of participants under 18 years of age.

# Results

## Flow of participants

S1 Fig shows the flow diagram of the study participants.

## Characteristics of study participants

The study involved 697 women of reproductive age, with a mean age of 29.7 ± 9.07 years.

Of the participants aged 18-49 years (n=634), 56.8% were employed. The proportion of individuals employed increased across age groups, from 33.7% among those aged 18-24 years to 75% among those aged 45-49 years (p<0.0001). Participants aged 18-49 years had a mean body mass index of 27.2 (±5.8). Among them, 1.9% were underweight (BMI of < 18.5 (underweight), while 28.9% were classified as obese (BMI of ≥ 30). The prevalence of obesity increased across age group and peaked at 39.3% among women aged 45-49 years of age. The differences in BMI distribution across the age groups were statistically significant (p<0.0001). The proportion of participants with a waist-to-height ratio of ≥ 0.5 (suggestive of abdominal obesity) increased with age group, from 14.1% among those 18-24 years old to 30.4% in the 45-49 years age group (p=0.003). The proportion of active individuals was highest among women aged 45-49 years (55.4%). In the 18-49 years, physical inactivity was prevalent (54.4%) and was more pronounced among younger women (p=0.019).

The prevalence of current alcohol use was 14.8%, with the highest proportion among women aged 35-44 years (20.3%). The variation across age groups was not statistically significant (p=0.061). The use of Tobacco was low (1.1%) among the 18-49-year-olds. The highest prevalence (3.1%) of Tobacco use was observed in the 35-44 age group (p=0.093). Most participants (82.0%) reported that they sometimes or never added salt to their meals. There were no significant differences regarding this salt use behaviour among the age groups (p=0.229) See Table 1.

## Age-specific evaluation of optimal thresholds for waist circumference using different classification metrics and prevalence of MetS components

For the age group 18-49 years, the optimal threshold was 80.3 cm. The Youden Index, specificity and sensitivity were 0.597, 65% and 94% respectively, and the AUC curve was 0.849 (95% CI 0.819 - 0.879).

The optimal WC cut-off points varied among the age groups, ranging from 79.9 cm for females aged 25 to 34 years, to 91.1 cm for those aged 45 to 49 years. Across different age

**Table 1. Characteristics of women of reproductive age in Wakiso district (N=634) stratified by age group.**

| Characteristics | Age group | | | | | P-value |
|---|---|---|---|---|---|---|
| All participants | 18-24 years (n=163) | 25-34 years (n=252) | 35-44 years (n=163) | 45-49 years (n=56) | 18-49 years (n=634) | |
| Age in years (mean, SD) | 21.2 ± 2.06 | 29.0 ± 2.7 | 38.8 ± 2.87 | 46.7 ± 1.70 | 31.1± 8.36 | p<0.0001[β] |
| Employment status (n, %) | | | | | | |
| Paid and employed | 55 (33.7) | 151 (59.9) | 112 (68.7) | 42 (75.0) | 360 (56.8) | p<0.0001 |
| Unpaid and unemployed | 108 (66.3) | 101 (40.1) | 51 (31.3) | 14 (25.0) | 274 (43.2) | |
| BMI[a] (kg/m²) (Mean, SD) | | | | | 27.2 ± 5.8 | |
| < 18.5 | 7 (4.3) | 2 (0.8) | 3 (1.9) | 0 (0.00) | 12 (1.9) | p<0.0001 |
| 18.5–24.9 | 93 (57.0) | 93 (36.9) | 53 (32.5) | 14 (25.0) | 253 (39.9) | |
| 25.0–29.9 | 35 (21.5) | 85 (33.7) | 46 (28.2) | 20 (35.71) | 186 (29.3) | |
| ≥ 30 | 28 (17.2) | 72 (28.6) | 61 (37.4) | 22 (39.3) | 183 (28.9) | |
| Waist height ratio (WHtR) (n, %) | | | | | | |
| < 0.5 | 140 (85.9) | 197 (78.2) | 117 (71.8) | 39 (69.6) | 493 (77.8) | 0.003 |
| ≥ 0.5 | 23 (14.1) | 55 (21.8) | 46 (28.2) | 17 (30.4) | 141 (22.2) | |
| Physical activity (n, %) | | | | | | |
| Active | 64 (39.3) | 119 (47.2) | 75 (46.0) | 31 (55.4) | 289 (45.6) | 0.019 |
| Inactive | 99 (60.7) | 133 (52.8) | 88 (54.0) | 25 (44.6) | 345 (54.4) | |
| Current alcohol users (Past 30 days) (n, %) | | | | | | |
| Current user | 17 (10.4) | 35 (13.9) | 33 (20.3) | 9 (16.1) | 94 (14.8) | 0.061 |
| Non-current drinker | 146 (89.6) | 217 (86.1) | 130 (79.7) | 47 (83.9) | 540 (85.2) | |
| Tobacco use (n, %) | | | | | | |
| Smokers (past 30 days) | 0 | 02 (0.8) | 05 (3.1) | 0 | 07 (1.1) | 0.093[α] |
| Non-smokers (past 30 days) | 163 (100.0) | 250 (99.2) | 158 (96.9) | 56 (100.0) | 627 (98.9) | |
| Dietary salt (n, %) | | | | | | |
| Adding salt or salty sauce at meal to food before or when eating | | | | | | |
| Sometimes - never | 129 (79.1) | 208 (82.5) | 132 (81.0) | 51 (91.1) | 520 (82.0) | 0.229 |
| Always or often | 34 (20.9) | 44 (17.5) | 31 (19.0) | 5 (8.9) | 114 (18.0) | |

*P. values were obtained using a one-way ANOVA for continuous variables (*[β]*) and a chi-square test for categorical variables.*

[α]*- Fisher exact test used*

groups, the area under the ROC curve at cut-point values fluctuated between 0.78 and 0.86, with the 45- 49-year-old age group showing the highest AUC of 0.86. The 45-49 years age group showed the highest discriminative power (AUC =0.891), with specificity (88%), Youden Index (0.729), and a likelihood ratio of (7.08).

The prevalence of various MetS components generally increased with age, with elevated waist circumference (p<0.001) and blood pressure (p<0.001) showing statistical significance. Reduced HDL cholesterol (p=0.958), elevated triglycerides (p=0.157) and elevated fasting blood glucose (p=0.572) did not show any significant differences across age groups as shown in Table 2.

## ROC curve analysis for waist circumference in reproductive-age women in Wakiso district for detecting MetS risk factors per age group

S2-S6 Figs show the ROC curves for WC to predict the presence of 2 or more MetS risk factors among females of reproductive age for the various age groups.

**Table 2. Age-stratified WC cut-off points, MetS risk classification metrics, and prevalence of MetS components by age group.**

| Variable | Age group | | | | |
|---|---|---|---|---|---|
| | 18-24 years (n=163) | 25-34 years (n=252) | 35-44 years (n=163) | 45-49 years (n=56) | 18-49 years (n=634) |
| **MetS risk Classification metrics** | | | | | |
| Empirical optimal WC cut-off point (cm) | 80.4 | 79.9 | 85.6 | 91.1 | 80.3 |
| Area under the ROC curve at the cut point | 0.84 | 0.78 | 0.80 | 0.86 | 0.80 |
| Area under ROC curve (95% CI) | 0.872 (0.817 - 0.927) | 0.804 (0.750 - 0.858) | 0.852 (0.786 - 0.918) | 0.891 (0.790 - 0.993) | 0.849 (0.819 - 0.879) |
| Sensitivity at cut point (%) | 86 | 97 | 85 | 85 | 94 |
| Specificity at cut point (%) | 82 | 58 | 75 | 88 | 65 |
| Youden index (J) | 0.688 | 0.557 | 0.604 | 0.729 | 0.597 |
| LR+ = sensitivity/ (1 - specificity) | 4.78 | 2.31 | 3.40 | 7.08 | 2.66 |
| Presence of 2 or more metabolic syndrome risk factors (%) | 40.5 | 46.8 | 63.2 | 69.6 | 51.4 |
| Waist height ratio (n, %) | | | | | |
| < 0.5 | 140 (85.9) | 197 (78.2) | 117 (71.8) | 39 (69.6) | 493 (77.8) |
| ≥ 0.5 | 23 (14.1) | 55 (21.8) | 46 (28.2) | 17 (30.4) | 141 (22.2) |
| **Metabolic syndrome components** | | | | | |
| Elevated waist circumference (%) [a] | 48.0 | 66.8 | 77.3 | 82.8 | 66.1 |
| Elevated triglycerides (%) [b] | 13.1 | 12.3 | 15.9 | 24.4 | 14.6 |
| Reduced HDL Cholesterol (%) [c] | 48.9 | 48.6 | 47.0 | 41.2 | 47.6 |
| Elevated blood pressure (%) [d] | 21.0 | 19.5 | 35.2 | 42.8 | 26.2 |
| Elevated fasting blood glucose (%) [e] | 11.7 | 5.0 | 6.8 | 6.7 | 7.4 |

[a]($\chi2=50.94$, df=4,p<0.001), [b]($\chi2=6.62$,df=4,p=0.157), [c]($\chi2=0.65$,df=4,p=0.958), [d]($\chi2=35.88$,df=4,p<0.001), [e]($\chi2=2.91$,df=4,p=0.572), ($\chi2=$ Pearson chi-square, df=degrees of freedom and p=p.value)

## Discussion

The rise in the global prevalence of MetS calls for more research towards early and accurate diagnosis of MetS to provide preventive measures and timely intervention for MetS.

Our study revealed that the mean waist circumference cut-off points of 80.3 cm accurately predicted MetS in females aged 18-49 years in an urban population in Uganda. However, optimal waist circumference varied by age group (18-24 years, 25-34 years, 35-44 years, and 45-49 years) being 80.4 cm, 79.9 cm, 85.6 cm, and 91.1 cm respectively. This study is the first to use the Joint Interim Statement to establish these cut-off thresholds in Uganda.

The findings showed a strong link between age and the prevalence of some MetS components, namely elevated waist circumference (p<0.001), and blood pressure (p<0.001), both of which increased with age group. Reduced HDL cholesterol (p=0.958), elevated triglycerides (p=0.157) and elevated fasting blood glucose (p=0.572) did not show any significant differences across age groups. Public health interventions for preventing metabolic syndrome should consider targeting abdominal obesity and hypertension, especially for older individuals within the reproductive age range. Research indicates that obesity [42] and hypertension [43] impact maternal and neonatal health.

Our study identified an optimal WC threshold value that closely approximates the recommended cut-off of 80.0 cm as specified by the International Diabetes Federation for diagnosing metabolic syndrome (MetS) [4]. Similar results were found in prior studies conducted across Africa and other locations. An optimal WC of 80.5 cm was found among female university

employees in Angola [44]. Elsewhere, some studies among a multiethnic Malaysian population [45] and in the Japanese population [46] found that 80.0 cm was an optimal WC for women. However, it should be noted that these two studies were conducted among ethnically different populations compared to Africans. The consistency of our findings with those of other researchers strengthens the validity of our findings and reinforces the global relevance of the established diagnostic criteria. From a public health perspective, policymakers and healthcare professionals can utilize these findings to implement cost-effective and efficient screening programs that facilitate the early detection and management of MetS.

However, our findings differed from a study in Ethiopia, which identified the optimal WC cutoff at 78.0 cm [47]. Studies conducted among women in Botswana and South Africa found higher optimal WC cut-off points of ≥ 82.3 cm and 92.0 cm, respectively [15,48]. These variations highlight the importance of considering region-specific factors while establishing WC cutoff points for assessing women's health. The observed variation might be attributed to differences in factors such as genetics, body composition, and socio-environmental factors across different geographical regions. For instance, countries with higher prevalences of obesity, such as South Africa, often have higher optimal WC cut-off points [15,49] due to different fat distribution. Economic conditions and urbanisation also affect lifestyle and dietary patterns [50], impacting obesity rates and waist circumference measurements.

This study also revealed varying age-specific WC thresholds for various age groups: 80.4 cm for people aged 18-24 years, 79.9 cm for those aged 25-34 years, 85.6 cm for the 35-44 years age group, and 91.1 cm for those aged 45-49 years, respectively. The age-specific cut-off points highlight the variability of MetS risk across various age groups. The younger individuals showed risk factors even with slightly lower waist circumference levels. This emphasizes the importance of early detection and timely intervention in this group. Conversely, elderly individuals exhibited higher waist circumference cut-off points. This is in agreement with other studies [51]. Probable reasons for this could be due to changes in body composition with age resulting in reduced muscle mass, gain in body fat, and a reduction in metabolic rate [52] which occurs with age resulting in weight gain if dietary intake and physical activity remained constant.

Therefore, to effectively combat the rising burden of cardiovascular diseases, healthcare providers and policymakers should include age-specific thresholds for waist circumference in screening and intervention protocols for metabolic syndrome.

## Diagnostic accuracy of the waist circumference cut-off points to detect MetS risk factors

Different metrics were employed to assess the precision of detecting metabolic syndrome, including AUC, PPV, specificity and sensitivity, with reference to a specific waist circumference threshold.

When we measured the performance of the AUC values within the 18-24 age range, the AUC reached 0.84. This data suggests that the WC cut-off point can serve as a reasonably accurate indicator of MetS risk in individuals within this age range. For the 25-34 years age group, the AUC was 0.78, this demonstrates moderate performance. In the 35-44 years age category, the AUC was 0.80, this suggests a relatively good ability to distinguish those at risk for MetS. In the 45-49 years age group, the AUC was 0.86. This age group exhibited the highest AUC values, and thus the most robust discriminative power. This suggests that this group may be the most appropriate target for MetS screening using WC. Considering all individuals aged 18-49 years, the overall AUC was 0.80, indicating good performance in identifying MetS. Normally, an AUC of 1 indicates a perfect classifier with the model yielding a 100% true

positive rate and 0% false positive rate across all possible threshold values. An AUC of 0 shows that the classifier performs no better than chance, is not able to distinguish between two classes and yields a 0% true positive rate and 100% false positive rate [53]. Most classifiers aim for an AUC above 0.5, and that indicates predictive power.

These findings hold important implications for healthcare professionals and researchers, as they provide valuable guidance for developing protocols for screening MetS that are specific to different age groups.

## Effectiveness of waist circumference cut-off points in terms of sensitivity and specificity

Considering sensitivity and specificity values, a WC of 80.4 cm in the 18-24 age group showed 86% sensitivity and 82% specificity. The high sensitivity further suggests its effectiveness in identifying metabolic syndrome.

The 25-34 age group showed a WC cut-off of 79.9 cm and had the highest sensitivity of 97%. This means that nearly all individuals with metabolic syndrome in this age group are accurately identified. However, a lower 58% specificity may imply concerns about potential false positives, and thus there is a need to do confirmation tests.

The 35-44 years age category had an 85.6 cm cut-off, and balanced sensitivity (85%) and specificity (75%). This implies that it is reasonably practical in identifying those with metabolic syndrome and can minimize the risk of false positives, making it a valuable tool for screening.

In the 45-49 age group, there was a 91.1 cm cut-off point, with an 85% sensitivity and 88% specificity. This implies that this cut-off can accurately identify individuals at risk and can minimise false positives. For the age range of 15-49 years, and 18-49 years, we obtained an 80.3 cm cut-off point, with 94% sensitivity and 67% specificity, and 94% sensitivity and 65% specificity, respectively. This shows that it has the potential to serve as an efficient screening tool. However, the lower specificity of 65% and 67%, respectively calls for further evaluation to confirm diagnoses.

Generally, these age-specific WC cut-off points can be used to offer tailored strategies for MetS risk assessment and highlight the importance of age in risk prediction and patient management. Further research may be needed to validate and expand these findings for broader clinical and public health applications.

## Youden Index and Positive Likelihood Ratios (LR+)

Our study evaluated age-specific waist circumference (WC) cut-off points and their associated Youden Index (J) and positive likelihood ratios (LR+) to determine the diagnostic performance across different age groups.

Our study identified variations among the different age categories. The 45-49-year-old age group showed the highest Youden Index (J) of 0.729, indicating a higher overall accuracy than the other age groups. A value of one (1) indicates perfect accuracy, and the closer J is to 1, the better the test's overall performance [54]. Furthermore, it showed the highest LR+ value of 7.08. This implies that among individuals within this age group, a positive test result is about 7.08 times more likely to occur in individuals with metabolic syndrome compared to those without the syndrome. This makes it a robust diagnostic tool.

The 25-34 years age group showed a slightly lower Youden Index (J) of 0.557 (indicating slightly lower overall accuracy) compared to the 18-24 years group with a Youden Index (J) of 0.688. The 25-34 years age group showed an LR+ of 2.3095, demonstrating valuable diagnostic utility. A LR+ of 2.3095 suggests that individuals diagnosed with metabolic syndrome have a

waist circumference above the cut-off point approximately 2.3 times more often than those without metabolic syndrome. Although this LR+ was slightly lower than the 18-24 years group LR+ (Likelihood Ratio Positive) of 4.788, this LR+ is still a valuable indicator of diagnostic strength. The 35 - 44 years and the 18 - 49 years age range both demonstrated intermediate Youden Index (J) values of 0.604 and 0.597, respectively, with corresponding LR+ values of 3.4 and 2.66.

The study demonstrates that for urban Ugandan females aged 18-49 years, 80.3 cm waist circumference is the mean optimal threshold for MetS prediction. However, these findings were based on data from one district and should be interpreted cautiously when generalizing to other populations. Furthermore, waist circumference cut-off points varied depending on age group as indicated; 80.4 cm (18-24 years), 79.9 cm (25-34 years), 85.6 cm (35-44 years), and 91.1 cm (45-49 years). Our findings provide useful information for healthcare professionals when considering age-tailored risk assessment strategies for metabolic syndrome to enhance the clinical relevance of WC measurements.

## Waist circumference cut-offs and Waist-to-Height Ratio (WHtR)

While our study determined optimal waist circumference cut-off points for predicting metabolic syndrome, assessing waist-to-height ratio (WHtR) could further enhance the interpretation of abdominal obesity and its associated risks. A waist-to-height ratio can account for variations in body height [55], making it a more standardised indicator of adiposity across diverse populations. This is particularly relevant in African populations, where regional differences in mean height may influence WC thresholds. For instance, a study that analysed cross-sectional data from Demographic and Health Surveys (1994-2008) in 54 low- to middle-income countries examined the height of women aged 25-49 years including in Africa. This study revealed variations in mean heights (cm) across different regions in Africa: 163.0± 6.7 cm in Senegal (West Africa), 162.6 ± 6.4 cm in Chad (Central Africa), 159.5 ± 6.0 cm in Egypt (North Africa), 156.6 ± 6.5 cm in Tanzania (East Africa), and 156.0 ± 6.2 cm in Mozambique (Southern Africa) [56]. Given that height can vary due to genetic, nutritional, and environmental factors, reliance on WC alone may not fully capture cardiometabolic risks across different subpopulations [57]. WHtR is a strong predictor of cardiometabolic risk and may provide more consistent cut-off points across age and ethnic groups [57]. Studies recommend a waist-to-height ratio threshold of 0.5 for identifying metabolic risk. A WHtR of ≥ 0.5, is indicative of increased risk, while < 0.5 is considered low risk for metabolic disorders [58]. Our study found an increase in the proportion of participants with a WHtR of ≥ 0.5 with the increase in age group, suggesting an increased cardiometabolic risk [58] as age increases.

Future research should assess the diagnostic accuracy of WHtR alongside WC to refine screening criteria for MetS in Uganda and other African settings so as to improve risk assessment in clinical and public health contexts.

## Study limitations

We used a cross-sectional study design, which captured WC at a single time point may not fully reflect the dynamic nature of MetS. A longitudinal study would appropriately confirm temporal relationships.

This study sample was randomly selected from an existing database, which may have introduced selection bias by excluding individuals not represented in the database. However, this bias was minimised as the sample was considered representative of Wakiso district.

Our sample size was calculated using MetS prevalence from a Kenyan urban female population. Although not identical to our study population, this limitation was minimised as our participants resided in an urban setting.

Variations in sample size across age groups may have affected the precision of some age-specific estimates. However, the use of sampling weights to adjust for selection probabilities and response rates likely improved the representativeness of the final sample. The weighted estimates may have accounted for differences in age distribution. Future studies may consider post-stratification weighting to further refine these estimates

The use of ≥ 2 components (excluding waist circumference) to define MetS, may overestimate its prevalence in the population. However, this facilitates early identification of individuals at metabolic risk and enhances detection and intervention efforts.

## Conclusions and recommendations

We found that a waist circumference of 80.3 cm was the optimal mean threshold for identifying metabolic syndrome in females between the ages of 18 and 49 years in Wakiso district, central Uganda. Furthermore, participants exhibited waist circumference cut-offs ranging from 79.9 cm to 91.1 cm based on age group. This study highlights the significance and precision of the existing waist circumference guidelines in healthcare. This study calls for the utilization of a waist circumference cut-off threshold to predict metabolic syndrome among reproductive-age females in Uganda, based on age group. The findings of this research may be relevant to healthcare providers, policymakers, public health practitioners and researchers. Further research in diverse populations, including both rural and urban settings is needed to validate these waist circumference cut-off points.

## Supporting information

**S1 Fig: Flow diagram of study participants**
(TIF)

**S2 Fig: ROC curve for 18-49 years old females in Wakiso district, 2023**
(TIF)

**S3 Fig: ROC curve for 18-24 years old females in Wakiso district, 2023**
(TIF)

**S4 Fig: ROC curve for 25-34 years old females in Wakiso district, 2023**
(TIF)

**S5 Fig: ROC curve for 35-44 years old females in Wakiso district, 2023**
(TIF)

**S6 Fig: ROC curve for 45-49 years old females in Wakiso district, 2023**
(TIF)

**S1 Data: Data for Optimal WC cutoff points**
(XLSX)

## Acknowledgements

We appreciate all the study participants and research assistants who supported the data collection process. We thank all the local leaders and the district health management team members in Wakiso district who facilitated the research teams. We appreciate Dr Arthur Bagonza (PhD) and Dr Freddie Kitutu (PhD) for reviewing the manuscript.

## Author contributions

**Conceptualization:** David Lubogo, Henry Wamani, Christopher Garimoi Orach.

**Data curation:** David Lubogo.

**Formal analysis:** David Lubogo, Roy William Mayega, Christopher Garimoi Orach.

**Funding acquisition:** David Lubogo, Christopher Garimoi Orach.

**Investigation:** David Lubogo, Henry Wamani, Christopher Garimoi Orach.

**Methodology:** David Lubogo, Henry Wamani, Roy William Mayega, Christopher Garimoi Orach.

**Project administration:** David Lubogo, Christopher Garimoi Orach.

**Resources:** David Lubogo, Roy William Mayega, Christopher Garimoi Orach.

**Software:** David Lubogo.

**Supervision:** David Lubogo, Henry Wamani, Christopher Garimoi Orach.

**Validation:** David Lubogo, Henry Wamani, Roy William Mayega, Christopher Garimoi Orach.

**Visualization:** David Lubogo.

**Writing – original draft:** David Lubogo.

**Writing – review & editing:** David Lubogo, Henry Wamani, Roy William Mayega, Christopher Garimoi Orach.

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
