## [Decision Letter · Decision Letter 0]

26 Aug 2024

PGPH-D-24-00473

Optimal waist circumference cut-off points for predicting Metabolic Syndrome among females of reproductive age in Wakiso district, central Uganda

Dear Dr. Lubogo,

Thank you for submitting your manuscript to PLOS Global Public Health. After careful consideration, we feel that it has merit but does not fully meet PLOS Global Public Health’s publication criteria as it currently stands. Therefore, we invite you to submit a revised version of the manuscript that addresses the points raised during the review process.

We look forward to receiving your revised manuscript.

Kind regards,

Sangeetha Shyam, M.Sc., PhD

Academic Editor

Journal Requirements:

1. Please update your online Competing Interests statement. If you have no competing interests to declare, please state: “The authors have declared that no competing interests exist.”

2. We note that your Data Availability Statement is currently as follows: "Data already provided as part of the submitted article."

Additional Editor Comments (if provided):

The manuscript aims at identifying an optimal waist circumference cut-off points for predicting Metabolic Syndrome in females of reproductive age in a district in central Uganda and makes a case for further larger studies in the country.

The authors are encouraged to conisder the comments from the reviewer.

Additionally while the results are obtained from one district in Uganda, the authors, Therefore temper statements such as these to indicate the appropriate level of certainity "The study demonstrates that for urban Ugandan females aged 15-49 years, 80.3 cm waist circumference is the mean optimal threshold for MetS prediction.

Reviewers' comments:

Reviewer's Responses to Questions

**Comments to the Author**

1. Does this manuscript meet PLOS Global Public Health’s publication criteria ? Is the manuscript technically sound, and do the data support the conclusions? The manuscript must describe methodologically and ethically rigorous research with conclusions that are appropriately drawn based on the data presented.

Reviewer #1: Partly

Reviewer #2: Yes

2. Has the statistical analysis been performed appropriately and rigorously?

Reviewer #1: No

Reviewer #2: Yes

3. Have the authors made all data underlying the findings in their manuscript fully available (please refer to the Data Availability Statement at the start of the manuscript PDF file)?

Reviewer #1: Yes

Reviewer #2: Yes

4. Is the manuscript presented in an intelligible fashion and written in standard English?

Reviewer #1: Yes

Reviewer #2: Yes

5. Review Comments to the Author

Reviewer #1: Review

Reviewer Recommendation and Comments for Manuscript Number PGPH-D-24-00473

Title: Optimal waist circumference cut-off points for predicting Metabolic Syndrome among females of reproductive age in Wakiso district, central Uganda.

Study aimed at determining the average waist circumference and age-specific cut-off points that are optimal for predicting MetS among females of reproductive age in Wakiso district, central Uganda. The hypothesis for the study was that the optimal waist circumference for females aged 15-49 years might not be equal to 80.0cm

My suggestion of definition:

INSERM/ Metabolic syndrome corresponds to the combination of several disorders related to the presence of excess fat inside the abdomen. Affected individuals have a large waist circumference (greater than 94 cm for men and 80 cm for women) and at least two other abnormalities among the following: hyperglycemia (excess blood sugar), elevated triglycerides, low "good" HDL cholesterol, and high blood pressure.

The International Diabetes Federation (IDF) has called on experts from around the world to provide a new definition of metabolic syndrome. According to the IDF, a person has metabolic syndrome when they have abdominal obesity (waist circumference greater than 94 cm in men and 80 cm in women) and at least two of the following factors:

• Elevated triglycerides: Triglyceride levels equal to or greater than 1.7 mmol/L, equivalent to 150 mg/dL.

• Low HDL cholesterol (the "good" cholesterol): HDL cholesterol levels less than 1.03 mmol/L (40 mg/dL) in men and less than 1.29 mmol/L (50 mg/dL) in women.

• High blood pressure: Blood pressure, also known as arterial pressure, equal to or greater than 130 mmHg systolic and 85 mmHg diastolic.

• Elevated fasting blood glucose: Fasting venous blood glucose equal to or greater than 5.6 mmol/L (100 mg/dL).

My remarks:

Introduction:

Problem: need to rework the structure of the introduction and the writing language. Avoid subjectivity and too much oral writing.

Suggestion:

I suggest the following outline:

• Definition and components of MetS

• Importance of early identification and management

• Significance of Waist Circumference (WC)

- WC as a measure of central obesity

- Comparison with Body Mass Index (BMI)

- WC's role in predicting MetS

• Current WC Cut-off Points

- Overview of existing guidelines and cut-off points (e.g., IDF recommendations)

- Variability of cut-off points based on population demographics

• Focus on Females of Reproductive Age

- Definition and characteristics of this demographic

- Impact of hormonal and physiological changes on metabolic health

- Unique metabolic challenges and risks in this group

• Research Gap

- Lack of specific studies on WC cut-off points for females of reproductive age

- Importance of identifying precise cut-off points for this population

• Study Objective

- Aim to determine optimal WC cut-off points for predicting MetS in females of reproductive age

- Potential benefits for early detection, prevention, and management of MetS

METHODS:

• Random Sampling from Database: While random sampling from a database ensures diversity, it may still miss out on segments of the population not included in the database, leading to potential selection bias.

• Sample Size Calculation: The sample size calculation assumes a response rate and prevalence based on previous studies from different populations, which may not accurately reflect the current study population's characteristics.

• Analysis: do we have data on sociodemographic informations?

•

RESULTATS:

• Flow-chart missing

• I would suggest to describe each component variables used to identify metabolic syndrome (WC, Elevated triglycerides, Low HDL cholesterol , Blood pressure, Elevated fasting blood glucose)

• We need the first table describing the study population, provide socio demographic information and also the variables related to the med syndrom.

• Then a table focusing on the waist circumference cut-off points and their 201 classification metrics for MetS risk

• Can we consider survival analysis

Given that your study is cross-sectional and aims to identify optimal waist circumference cut-off points for predicting Metabolic Syndrome (MetS) among females of reproductive age, the recommended modeling analyses would primarily focus on classification and predictive modeling. Here are some suitable methods:

• Logistic Regression Analysis:

- To assess the relationship between waist circumference and the presence of MetS, while controlling for other potential confounders (e.g., age, BMI, lifestyle factors).

- Output: Odds ratios (ORs) indicating the likelihood of MetS associated with different waist circumference values.

- Application: Logistic regression can help determine the optimal cut-off point for waist circumference by evaluating the sensitivity and specificity at various thresholds.

• Receiver Operating Characteristic (ROC) Curve Analysis:

- Purpose: To evaluate the diagnostic performance of waist circumference in predicting MetS.

- Output: Area under the curve (AUC) to measure the overall accuracy. The optimal cut-off point can be identified using the Youden Index (sensitivity + specificity - 1).

- Application: ROC analysis will help in identifying the waist circumference value that maximizes both sensitivity and specificity for predicting MetS.

• Sensitivity and Specificity Analysis:

- Purpose: To calculate the sensitivity, specificity, positive predictive value (PPV), and negative predictive value (NPV) for various waist circumference cut-off points.

- Application: This analysis will provide detailed information on how well different cut-off points classify individuals as having or not having MetS.

• Youden Index Calculation:

- Purpose: To find the cut-off point that maximizes the Youden Index, which is a summary measure of the ROC curve.

- Application: The Youden Index helps to identify the threshold that offers the best trade-off between sensitivity and specificity.

Reviewer #2: Dear Author,

Thank you for the opportunity to review your manuscript. After careful consideration, I have several suggestions that may improve the clarity and impact of your work:

I have attached my comments in the Word document; please refer to it for detailed feedback.

Ethical Considerations: I did not find any issues with dual publication or research ethics.

Overall Quality: Your manuscript is well-structured and addresses an important topic. With these revisions, it has the potential to contribute significantly to the field.

I look forward to seeing the revised manuscript.

6. PLOS authors have the option to publish the peer review history of their article (what does this mean? ). If published, this will include your full peer review and any attached files.

**Do you want your identity to be public for this peer review?** For information about this choice, including consent withdrawal, please see our Privacy Policy .

Reviewer #1: No

Reviewer #2: **Yes: ** Dr.Lakshmi Priya Nagarajan

---

## [Decision Letter · Decision Letter 1]

3 Dec 2024

PGPH-D-24-00473R1

Optimal waist circumference cut-off points for predicting Metabolic Syndrome among females of reproductive age in Wakiso district, central Uganda

Dear Dr. Lubogo,

Thank you for submitting your manuscript to PLOS Global Public Health. After careful consideration, we feel that it has merit but does not fully meet PLOS Global Public Health’s publication criteria as it currently stands. Therefore, we invite you to submit a revised version of the manuscript that addresses the points raised during the review process.

We look forward to receiving your revised manuscript.

Kind regards,

Sangeetha Shyam, M.Sc., PhD

Academic Editor

Journal Requirements:

Additional Editor Comments (if provided):

The authors are commended for addressing the reviewer comments in this version of the manuscript. They are encouraged to address the minor comments from the reviewer

Reviewers' comments:

Reviewer's Responses to Questions

**Comments to the Author**

1. If the authors have adequately addressed your comments raised in a previous round of review and you feel that this manuscript is now acceptable for publication, you may indicate that here to bypass the “Comments to the Author” section, enter your conflict of interest statement in the “Confidential to Editor” section, and submit your "Accept" recommendation.

Reviewer #2: All comments have been addressed

2. Does this manuscript meet PLOS Global Public Health’s publication criteria ? Is the manuscript technically sound, and do the data support the conclusions? The manuscript must describe methodologically and ethically rigorous research with conclusions that are appropriately drawn based on the data presented.

Reviewer #2: Yes

3. Has the statistical analysis been performed appropriately and rigorously?

Reviewer #2: Yes

4. Have the authors made all data underlying the findings in their manuscript fully available (please refer to the Data Availability Statement at the start of the manuscript PDF file)?

Reviewer #2: Yes

5. Is the manuscript presented in an intelligible fashion and written in standard English?

Reviewer #2: Yes

6. Review Comments to the Author

Reviewer #2: Dear Author

Thank you for the opportunity to review your manuscript on MetS prevalence and WC as a predictor. I commend your work and have a few suggestions for improving clarity.

• BMI Classification: Use standard categories like WHO’s cut-offs. However, since adolescents are analyzed in Table 1, the BMI categorization should be done separately for adolescents based on age- and sex-specific percentiles (WHO guidelines) or remove adolescent throughout analysis [because I also saw in table 2 results that reveal that for the 15-17 age group, the AUC value of 0.81 indicates good performance, but the broad confidence interval (0.690-0.937) may be due to the small sample size (n=63). Sensitivity (87%) is high, but specificity (75%) is moderate, and the Youden index (0.620) is relatively low. larger sample sizes in future studies could improve the results.

• Physical Activity Levels: Specify the classification method, e.g., WHO guidelines, PAL, or GPAQ.

• Alcohol Use Levels: Define low, intermediate, and high levels, and mention the measurement standard used.

These comments are also highlighted in the attached draft.

Final Remarks

With these updates, the paper will be clearer and more informative. Thank you for considering these suggestions. I look forward to the impact of your research.

Best regards,

7. PLOS authors have the option to publish the peer review history of their article (what does this mean? ). If published, this will include your full peer review and any attached files.

**Do you want your identity to be public for this peer review?** For information about this choice, including consent withdrawal, please see our Privacy Policy .

Reviewer #2: **Yes: ** Lakshmi priya Nagarajan

---

## [Decision Letter · Decision Letter 2]

20 Jan 2025

PGPH-D-24-00473R2

Optimal waist circumference cut-off points for predicting metabolic syndrome among females of reproductive age in Wakiso district, central Uganda

Dear Dr. Lubogo,

Thank you for submitting your manuscript to PLOS Global Public Health. After careful consideration, we feel that it has merit but does not fully meet PLOS Global Public Health’s publication criteria as it currently stands. Therefore, we invite you to submit a revised version of the manuscript that addresses the points raised during the review process.

We look forward to receiving your revised manuscript.

Kind regards,

Sangeetha Shyam, M.Sc., PhD

Academic Editor

Journal Requirements:

Additional Editor Comments (if provided):

The authors are commended for addressing the previous comments from the reviewers.

Kindly consider the minor comments from the reviewer.

Reviewers' comments:

Reviewer's Responses to Questions

**Comments to the Author**

1. If the authors have adequately addressed your comments raised in a previous round of review and you feel that this manuscript is now acceptable for publication, you may indicate that here to bypass the “Comments to the Author” section, enter your conflict of interest statement in the “Confidential to Editor” section, and submit your "Accept" recommendation.

Reviewer #2: All comments have been addressed

2. Does this manuscript meet PLOS Global Public Health’s publication criteria ? Is the manuscript technically sound, and do the data support the conclusions? The manuscript must describe methodologically and ethically rigorous research with conclusions that are appropriately drawn based on the data presented.

Reviewer #2: Partly

3. Has the statistical analysis been performed appropriately and rigorously?

Reviewer #2: Yes

4. Have the authors made all data underlying the findings in their manuscript fully available (please refer to the Data Availability Statement at the start of the manuscript PDF file)?

Reviewer #2: Yes

5. Is the manuscript presented in an intelligible fashion and written in standard English?

Reviewer #2: Yes

6. Review Comments to the Author

Reviewer #2: Dear Author

We appreciate your effort and commend the valuable research you have conducted in a low- and middle-income country on optimal waist circumference cut-off points for predicting metabolic syndrome among females of reproductive age in Wakiso district, central Uganda, contributing significantly to the global understanding of this important health issue. Your work highlights findings that can inform public health policies and interventions.

We believe the manuscript can be further strengthened by enhancing its scientific clarity and depth of analysis. Specifically:

Strengthening the Analysis: Incorporating detailed association analyses while accounting for potential confounding factors, particularly by age stratification, would provide stronger support for the conclusions.

Broader Anthropometric Insights: Including waist-to-height ratio (WHtR) as an additional measure and discussing its relevance in the context of regional variations in mean height within African populations will enhance the interpretation of central obesity and related risks in the discussion part.

Clarifying Context: Providing a thoughtful discussion of the study’s scope and constraints, framed with supporting evidence, will help improve the manuscript's scientific impact.

We appreciate your valuable contribution to this field of research and encourage you to consider these suggestions to further enhance your manuscript.

I have attached my detailed comments directly within the manuscript for your reference.

Best wishes for your continued research endeavors.

Kind regards

7. PLOS authors have the option to publish the peer review history of their article (what does this mean? ). If published, this will include your full peer review and any attached files.

**Do you want your identity to be public for this peer review?** For information about this choice, including consent withdrawal, please see our Privacy Policy .

Reviewer #2: **Yes: ** Dr Lakshmi priya Nagarajan

---

## [Editor Report · Decision Letter 3]

11 Mar 2025

Optimal waist circumference cut-off points for predicting metabolic syndrome among females of reproductive age in Wakiso district, central Uganda

PGPH-D-24-00473R3

Dear Dr Lubogo,

We are pleased to inform you that your manuscript 'Optimal waist circumference cut-off points for predicting metabolic syndrome among females of reproductive age in Wakiso district, central Uganda' has been provisionally accepted for publication in PLOS Global Public Health.

Best regards,

Sangeetha Shyam, M.Sc., PhD

Academic Editor